# Influence of Hot Consolidation Conditions and Cr-Alloying on Microstructure and Creep in New-Generation ODS Alloy at 1100 °C

**DOI:** 10.3390/ma13225070

**Published:** 2020-11-10

**Authors:** Jiří Svoboda, Natália Luptáková, Milan Jarý, Petr Dymáček

**Affiliations:** Institute of Physics of Materials of the CAS, 616 62 Brno, Czech Republic; luptakova@ipm.cz (N.L.); jary@ipm.cz (M.J.); pdymacek@ipm.cz (P.D.)

**Keywords:** creep, ODS alloy, mechanical alloying, consolidation via hot rolling, microstructure

## Abstract

The coarse-grained new-generation Fe-Al-Y_2_O_3_-based oxide dispersion strengthened (ODS) alloys contain 5 vol.% homogeneously dispersed yttria nano-precipitates and exhibit very promising creep and oxidation resistance above 1000 °C. The alloy is prepared by the consolidation of mechanically alloyed powders via hot rolling followed by secondary recrystallization. The paper presents a systematic study of influence of rolling temperature on final microstructure and creep at 1100 °C for two grades (Fe-10Al-4Y_2_O_3_ and Fe-9Al-14Cr-4Y_2_O_3_ in wt%) of new-generation ODS alloys. The hot rolling temperatures exhibit a rather wide processing window and the influence of Cr-alloying on creep properties is evaluated as only slightly positive.

## 1. Introduction

The development of creep and oxidation-resistant alloys for applications at very high temperatures up to 1300 °C is still a current challenge due to the necessity of increasing of efficiency and reliability of high temperature testing machines and energetic equipment. The Ni-based superalloys are strengthened by coherent ordered γ’-precipitates. The mechanisms of creep in single crystals are summarized in the overview paper [1] and addressed by micromechanical unit cell model [2]. However, the γ’-phase cutting by dislocations and the instability of the γ’-precipitates due to their coarsening or rafting above 900 °C set strength and creep time limits for the superalloys. Thus, the temperature of 1100 °C is the upper bound for long-term applications of Ni-based superalloys. Moreover, enormous processing costs and complicated machinability seem to be the greatest drawback of the Ni-based superalloy single crystals. In contrast, the tungsten heavy alloys [3] exhibit a significant creep resistance up to 1500 °C; however, their oxidation resistance is rather poor.

The oxide dispersion strengthened (ODS) ferritic steels (alloys) containing a sufficient amount of Al seem to be the best solution to the problem, as the yttria nano-dispersion exhibits a significant resistance against coarsening and the Al content guarantees an excellent oxidation resistance in the temperature regime of 1100–1300 °C [4]. The microstructure of classical ODS alloys consists of the matrix strengthened by the dispersion of Y-Ti-Al-based nanooxides of typical size between 5 and 30 nm and of a volume fraction of about 0.5%. The recent commercial ODS alloys are represented by MA956 or MA957 [5], PM 2000 or PM 2010 [6], ODM alloys [7], 1DK or 1DS [8]. The non-commercial, experimental, and advanced versions of ODS alloys are ODS Eurofer [9], 12YWT [10], 14YWT [11], and 9YWT [12]. The mechanical properties of ODS alloys are currently improved via optimized processing, which focuses on a controlled thermo-mechanical treatment and on the suppression of nitrogen and carbon pick-up during processing [13]. The excellent creep strength of the ODS alloys is associated with an attractive interaction between dislocations and nanooxides as it was successfully modeled by Rösler and Arzt [14], who also predicted a threshold stress for creep and its significant drop with coarsening of the nanooxides. The coarsening kinetics of particles in multicomponent systems is modeled by Fisher et al. [15] and experimentally studied e.g., by Bartkova et al. [16] or by Svoboda et al. [17], indicating a very high resistance of Y-nanooxides against coarsening in ODS alloys. However, the high-temperature applications of ODS alloys are limited by the existence of grain boundaries that represent paths of high diffusivity and locations of intergranular damage. Thus, an optimum design of top creep-resistant ODS alloys must aim at rather coarse grains with a sufficient intergranular cohesion strengthened by a homogeneous dispersion of stable nanooxides of a sufficient volume fraction.

ODS alloys are typically produced in two steps. Very homogeneous powder consisting of the matrix and nano-sized Y_2_O_3_ particles is produced by mechanical alloying (MA) and consolidated via hot extrusion, hot isostatic pressing, or spark plasma sintering [11,12,18,19,20,21]. Additional thermomechanical processing is often applied to ensure the elimination of porosity and enhance their performance. Recently, hot cross rolling was used by Auger et al. [22] to improve the resistance of 14YWT ODS alloy against radiation. Zhang et al. [23] applied hot rolling to increase the strength of Fe-9Cr-0.06C-1.5W-0.5Ti-0.18Si-0.35Y_2_O_3_ ODS alloy (wt% are used in all notations). Kumar et al. [24] studied the mechanical properties of Fe-18Cr-2W-0.2Ti-xY_2_O_3_ alloy consolidated from mechanically alloyed powder by forging. Zhou et al. [25] applied combinations of forging, hot rolling, cold rolling, and annealing to ODS310 alloy with Mo addition. However, inhomogeneous grain size distribution and brittle cracking along the grain boundaries are still issues for the ODS alloys [26]. Although the coarse grain microstructure achieved by completed secondary recrystallization seems to be the key factor for acquiring excellent creep strength, several ODS alloys presented in the available literature have not met this requirement yet [26]. Just finding a simple processing route leading reliably to the coarse grained microstructure seems to be the key task in the research of the ODS alloys.

Very recently, the authors have finished the development of new-generation ODS alloys after several years of experimental work in their institute. The basic differences between the new-generation and classical ODS alloys are by one order of magnitude higher volume fraction of pure Y nanooxides, the strict requirement of coarse-grained microstructure, and significantly simplified processing. Excellent creep and oxidation resistances are obtained for a simple Fe-10Al-4Y_2_O_3_ system, as shown in this paper and [4], while the influence of alloying by Ti, Mo, Ni, Ta, and Co is evaluated as negative or neutral (unpublished results). According to our observations, it has been found that decreasing the Al content leads to degradation of both oxidation and creep resistances and the content of about 10% of Al is optimal, still keeping ferritic matric and minimizing density.

The aim of the present paper is to present the microstructure and creep properties of new-generation ODS alloys and show the influence of the basic processing parameter (rolling temperature during consolidation) on the grain microstructure and creep properties. The influence of Cr alloying on the microstructure and creep properties is also evaluated.

## 2. Materials and Methods

### 2.1. Material Processing

The new-generation ODS alloys of the chemical compositions Fe-10Al-4Y_2_O_3_ and Fe-9Al-14Cr-4Y_2_O_3_ are produced in three processing steps:The ODS powders are prepared from the powders of individual components in a self-made ball mill by MA. A vacuum-tight milling container with a volume of 22 dm^3^ and diameter of 400 mm made from low alloyed steel is filled with 100 bearing balls of diameter 40 mm (25 kg). The total amount of 1 kg of the powder is mechanically alloyed by rotation of the milling container at 70 rpm along the horizontal axis. After a sufficiently long MA of the powder in vacuum (two weeks), its properties become saturated, and the powder particles consist of a homogeneous solid solution with a huge density of defects such as dislocations and vacancies. This ensures also the complete dissolution of input yttria powder in the mechanically alloyed powder; see [27] for the respective theoretical reasoning. A larger amount of powder, e.g., 2.5 kg, can also be used if the MA time is proportionally increased. A small amount of Cr (0.05 wt%—determined by Energy-dispersive X-ray spectrometer (EDX) analysis) in the Fe-10Al-4Y_2_O_3_ powder after MA indicates that about 5% of the powder originates from the continuous abrasion of the milling balls made from Fe-1Cr-1C bearing steel. This amount of abrasion determines also the amount of C (about 0.05 wt%) in the new-generation ODS alloy.After MA, the powder is poured and cold compacted into a rolling container from a low-alloyed steel tube with 20 mm diameter and 1 mm wall thickness. Then, the rolling container is evacuated, sealed by welding, and rolled at temperatures in the range of 840–1020 °C in three steps to the thicknesses of 7.5, 4.9, and 3.25 mm. The mean strain rate during hot rolling is estimated to be 10 s^−1^. An intensive dynamic recrystallization occurring during hot rolling leads to an ultra fine-grained pore-free microstructure (see [16] and Section 3 in this paper).After the hot consolidation, the new-generation ODS alloy is stripped from the rolling container and annealed at 1200 °C for 4 h to achieve the required coarse-grained microstructure with homogeneously dispersed nano-oxides by secondary recrystallization. The kinetics of microstructure evolution after hot rolling in similar systems is analyzed in [16,17].

### 2.2. Microstructure Characterization

The microstructure characterization of the specimens after hot rolling, secondary recrystallization, and creep is performed using a scanning electron microscope (SEM) Tescan LYRA 3 XMU FEG/SEMxFIB (Brno, Czech Republic) analyzing the material in backscattered-electron (BSE) mode. Metallographic samples are prepared by mechanical–chemical polishing using a colloidal silica emulsion. The plain of the metallographic sections is normal to the rolling direction in the case of microstructures after hot rolling and after secondary recrystallization. In the case of microstructure characterization after creep tests, the metallographic section is parallel to the rolling direction and normal to the rolling plane.

### 2.3. Hardness Measurement

The hardness of the coarse-grained microstructure after annealing is measured using a Vickers indenter loaded with a force corresponding to 5 kg weight (ZwickRoell ZHV30 Vickers Hardness Tester) (Fürstenfeld, Austria).

### 2.4. Specimens Preparation

For the tensile tests and creep tests, flat dog-bone-shaped specimens are used. The axis of specimens shaped by spark erosion is parallel to the rolling direction (see Figure 1). The final specimen thickness of 2.4 mm is achieved by grinding.

### 2.5. Loading of Specimens

The specimens are loaded in the direction parallel to the rolling direction. The tensile tests at low strain rate 10^−6^ s^−1^ are performed using a Zwick/Roell KAPPA LA (Fürstenfeld, Austria) spring 50 kN universal creep machine at 1100 °C. The tensile tests allow quick estimation of the tensile/creep strength depending on the rolling temperature and selection of the most promising heats for the creep testing. The creep tests on air in the range of stress 55–85 MPa at 1100 °C are performed using direct-load 1.5 kN creep machines of in-house design. The deformation of specimen during the creep test is measured indirectly using the Micro-Epsilon triangular laser sensor located on the pull rod assembly near the deadweight.

## 3. Results and Discussion

The microstructures of the material under investigation are shown in Figure 2, Figure 3 and Figure 4. The typical ultra fine-grained microstructure after consolidation of the new-generation ODS alloys by hot rolling is shown in Figure 2a. An intensive dynamic recrystallization occurs during hot rolling, which leads to the observed nearly equi-axed ultra fine-grained microstructure. Moreover, compared to the initial state of the powder after MA, the density of defects significantly decreases by dynamic recrystallization, oxygen trapped at dislocations is released into the lattice, and an intensive precipitation of Y nanooxides occurs. They are detected by transmission electron microscopy (TEM, JEOL JEM-2100F, JEOL, Tokyo, Japan), and their mean size is about 5 nm; see Figure 1b in reference [28] for the detailed TEM microstructure, which cannot be resolved in Figure 2a. During secondary recrystallization, the nanooxides coarsen predominantly by fast diffusion along grain boundaries in the ultra fine-grained microstructure pinned by the nanooxides. After the precipitates coarsen sufficiently to the size of about 20 nm, the pinning effect becomes insufficient, and the secondary recrystallization starts to take place; see [28] for an explanation of the thermodynamic model. After the secondary recrystallization, the nanooxides in large grains (see Figure 2b) in the grain interiors (separated from the grain boundaries) become much more resistant against coarsening. From Figure 2b, one can estimate the mean size of the precipitates to be 20 nm, and their volume fraction of 5% follows from the chemical composition and the assumption that all oxygen is bonded in the yttrium oxides.

The grain microstructures after secondary recrystallization at 1200 °C/4 h of the Fe-10Al-4Y_2_O_3_ new-generation ODS alloys consolidated by rolling at temperatures 840 °C, 900 °C, 960 °C, and 1020 °C are presented in Figure 3, and those of Fe-9Al-14Cr-4Y_2_O_3_ new-generation ODS alloys consolidated by rolling at the same temperatures are presented in Figure 4. From Figure 3, one can conclude that the mean size and morphology of secondary recrystallized grains in the Fe-10Al-4Y_2_O_3_ new-generation ODS alloy are practically the same in specimens rolled at temperatures of 840 °C, 900 °C, and 960 °C (mean size typically 100–150 μm), while the grains in specimen rolled at temperature 1020 °C are significantly coarser (mean size 300–500 μm). It is necessary to note that the shape of the grains is rather irregular, and thus, a precise measurement of the grain size is subjected to a high systematic error. A different conclusion can be made for the Fe-9Al-14Cr-4Y_2_O_3_ new-generation ODS alloys shown in Figure 4, where the secondary recrystallized grains are much coarser in specimens rolled at temperatures of 900 °C and 960 °C (mean size typically 200–400 μm) than in those rolled at temperatures of 840 °C and 1020 °C (mean size 80–150 μm). The texture formation has been analyzed for the very similar system in [29], and no significant texture formation has been observed.

Several Cr-rich carbides precipitated predominantly at grain boundaries in Fe-9Al-14Cr-4Y_2_O_3_ new-generation ODS alloys are identified by EDX as mixed (Fe,Cr)_3_C carbides (see μm sized particles in Figure 4). These carbides precipitate during slow cooling in the vacuum furnace after secondary recrystallization treatment. If the specimens are heated up to 1100 °C and cooled within one minute, the carbides disappear. Thus, one can conclude that the carbides dissolve at temperatures above 1000 °C and cannot influence the properties at such temperatures and higher ones. In other words, the amount of 0.05% of C has no negative influence on high-temperature properties.

The results of HV5 hardness measurements are summarized in Figure 5a. The correlation between the hot rolling conditions and the grain size is insignificant. In contrary, the increase of the hardness due to Cr alloying is much more pronounced (the increase is by about 40 Vickers). The results of tensile testing are summarized in Figure 5b,c. The correlation between the hot rolling conditions and the tensile strength (see Figure 5b) is significant and indicates that the optimal rolling temperatures are between 870 °C and 960 °C for the Fe-10Al-4Y_2_O_3_ alloy and between 900 °C and 960 °C for the Fe-9Al-14Cr-4Y_2_O_3_ alloy. There is no significant correlation between the hot rolling conditions and the ductility (see Figure 5c). These results clearly indicate that the hot consolidation by rolling has a relatively wide processing window for temperatures between 900 °C and 960 °C, which allows for a very good and easy manufacturing reproducibility of new-generation ODS alloys. The relatively low value of the ductility is discussed later in connection with results of creep tests.

Based on the microstructural characterization and results of the tensile tests, the most promising heats Fe-10Al-4Y_2_O_3_ rolled at 900 °C and Fe-9Al-14Cr-4Y_2_O_3_ rolled at 930 °C have been selected for the creep study. The results of creep tests at 1100 °C are summarized in Figure 6. The times to fracture in dependence on applied stress are presented in Figure 6a, the secondary creep rates in dependence on applied stress are presented in Figure 6b, and the stress dependence of ductility is presented in Figure 6c. The times to fracture are also compared with those of the top commercial ODS alloy MA956 [5] in Figure 6a. The longest time to fracture (2329 h), the lowest secondary creep rate (2.3 × 10^−9^ s^−1^), and the highest ductility (2.25%) are observed for the heat Fe-10Al-4Y_2_O_3_ crept at 55 MPa. The respective creep curve together with the longest creep curves of other heats are presented in Figure 6d. It should be pointed out that the creep properties of new-generation ODS alloys demonstrated by the creep curve in Figure 6d are very suitable for the design of structures requiring excellent oxidation resistance as well as long-term shape and dimensional stability under mechanical loading at very high temperatures. The extent of the primary creep (0.4%) and the fast onset of a very low secondary creep strain rate are rather exceptional, while ductility is fully sufficient (1% of plastic strain is usually considered as an acceptable limit for deformation in structures). As the absolute values of the stress exponents of time to fractures as well as of the secondary creep rate are very high and a significant increase of ductility occurs by decreasing the applied stress, amazing design features of the new generation ODS alloys can be expected at loading corresponding to stress levels below 40 MPa.

With respect to the very promising applicability of the new-generation ODS alloys, it is highly desirable to study the fracture mechanism predominantly for long-term creep tests. As stated in the Introduction, one of the limiting factors of the strength is the cohesive strength of grain boundaries, which is exhibited by rather brittle intergranular fracture. However, if the applied stress is decreased, the crack propagation along the grain boundaries becomes more and more conditioned by stress redistribution in the creeping specimen and by the development and growth of defects such as creep cavities. As the stress exponent of diffusive cavity growth is quite low near 1, see [30], the most pronounced cavitation can be expected in specimens with the highest time to fracture, i.e., in the heat Fe-10Al-4Y_2_O_3_ crept at 55 MPa. The fracture mechanism is analyzed in Figure 7, showing the intergranular creep fracture (Figure 7a,b), the metallographic cross-section in the specimen head (Figure 7c), and the metallographic cross-section in the specimen bulk near to the fracture surface (Figure 7d).

In Figure 7a, one can observe the intergranular brittle fracture along the grain boundaries of pancake grains. The detail of the fracture surface (see Figure 7b) indicates that the brittle fracture is initiated at intergranular cavities. As shown in Figure 7c, the cavities are present also in the head of the specimen, and the same cavitation is also indicated in the virgin specimen after secondary recrystallization. From that, one can deduce that the cavities are represented by bubbles of gas trapped in the hot consolidated material and the gas precipitates as bubbles during secondary recrystallization. This hypothesis is supported by the fact that the bubbles are situated predominantly near the center of the specimen, because the trapped gas can diffuse out of the specimen near the surface. By a comparison of Figure 7c,d, one can conclude that no new cavities are nucleated in the specimen during creep and that the cavities grow very slowly. It is the challenge for the authors to find the reason for the formation of bubbles during secondary recrystallization and prohibit their formation. Most probably, the degassing of the powder will prohibit the bubbles formation, which will be the topic of future studies.

## 4. Conclusions and Outlook

This paper featured high-temperature tensile tests and creep tests at 1100 °C completed by detailed microstructure characterization and hardness measurements of the new-generation Fe-10Al-4Y_2_O_3_ and Fe-9Al-14Cr-4Y_2_O_3_ ODS alloys prepared by the rolling of mechanically alloyed powders at temperatures ranging from 840 °C to 1020 °C followed by annealing at 1200 °C/4 h. The conclusions can be summarized in the following items:The microstructure of the new-generation ODS alloys consists of large grains (80–400 μm, not much depending on chemical composition and rolling temperature) strengthened by the homogeneous dispersion of yttria nano-precipitates (20 nm) of the volume fraction of 5%.The results of microstructure characterization and of mechanical testing indicate that a rather wide processing window of rolling temperatures from 900 °C to 960 °C exists to provide very good creep properties of the new-generation ODS alloys at 1100 °C. The creep strength exceeds that of the top commercial ODS alloy MA956 by about 30%.The influence of Cr content in the system does not seem to play an important role in both mechanical properties and grain microstructure. The presence of Cr causes the formation of (Fe,Cr)_3_C carbides; however, these dissolve at the testing temperature 1100 °C, and thus, they have no influence on creep properties.The grain boundary decohesion mechanism initiated at cavities is detected to be responsible for the rather brittle creep fracture of the new-generation ODS alloys. However, the ductility significantly increases with decreasing applied stress, which is a promising message for practice utilizing substantially lower applied stresses than those used in the present study.

The experiments performed in this paper approve very promising creep properties of the new-generation ODS alloys at 1100 °C. As the oxidation resistance and stability of oxides against coarsening has potential up to 1300 °C, this motivates performing further mechanical and microstructural studies for temperatures of 1200 °C and 1300 °C in the nearest future.

## Figures and Tables

**Figure 1 materials-13-05070-f001:**
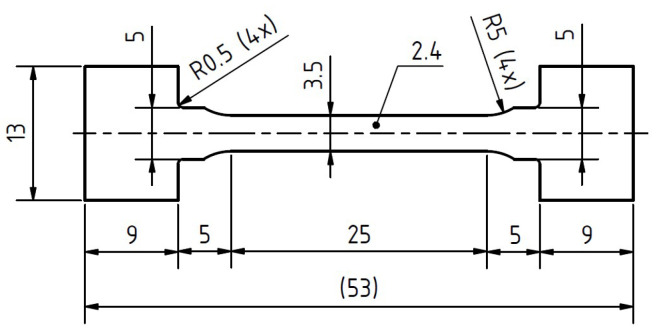
Dimensions and shape of the flat dog-bone specimens for tensile and creep testing.

**Figure 2 materials-13-05070-f002:**
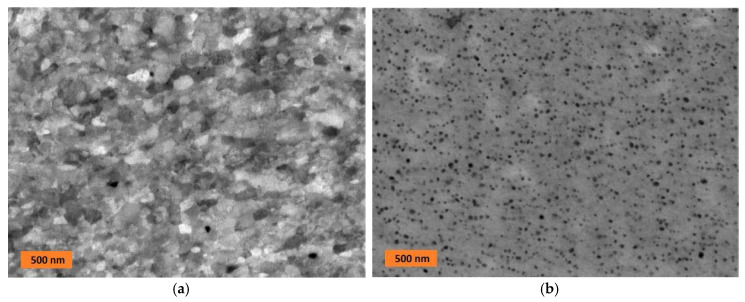
SEM micrograph (in backscattered electrons (BSE)) of (**a**) the typical ultra fine-grained microstructure of the new-generation oxide dispersion strengthened (ODS) alloy after hot rolling and (**b**) detail of the typical microstructure of nanooxide dispersion in large grains after secondary recrystallization.

**Figure 3 materials-13-05070-f003:**
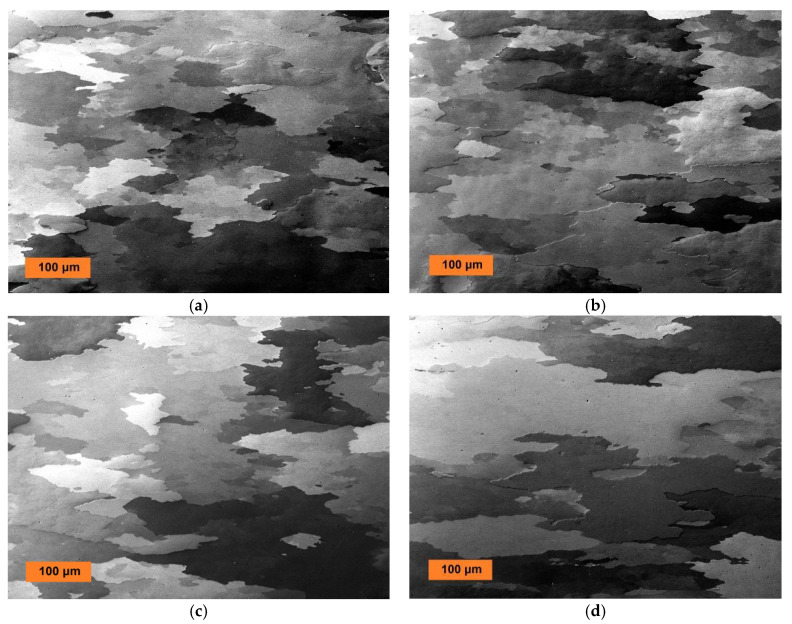
SEM micrograph (in BSE) of the coarse-grained microstructures of Fe-10Al-4Y_2_O_3_ new-generation ODS alloy after secondary recrystallization hot rolled at (**a**) 840 °C, (**b**) 900 °C, (**c**) 960 °C, and (**d**) 1020 °C.

**Figure 4 materials-13-05070-f004:**
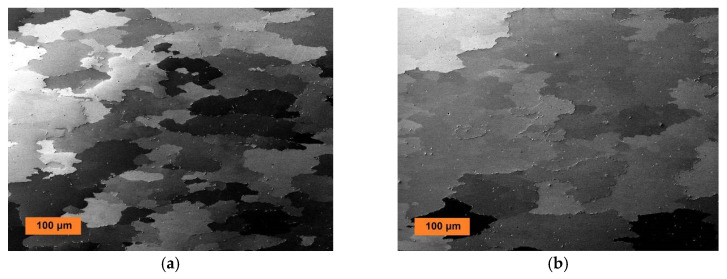
SEM micrograph (in BSE) of the coarse-grained microstructures of Fe-9Al-14Cr-4Y_2_O_3_ new-generation ODS alloy after secondary recrystallization hot rolled at (**a**) 840 °C, (**b**) 900 °C, (**c**) 960 °C, and (**d**) 1020 °C.

**Figure 5 materials-13-05070-f005:**
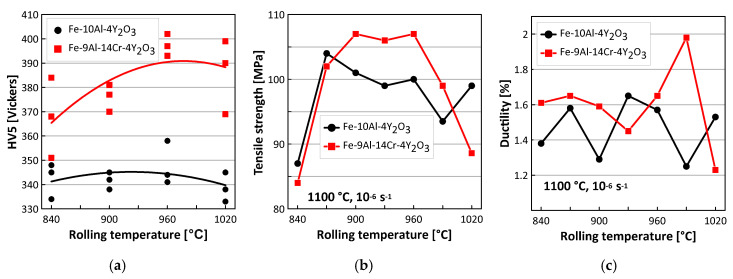
Summarizing results of influence of rolling temperature on (**a**) hardness measurements, and on (**b**) strength and (**c**) ductility determined by tensile testing at strain rate $\dot{\varepsilon }=10^{-6}s^{-1}$ and temperature 1100 °C.

**Figure 6 materials-13-05070-f006:**
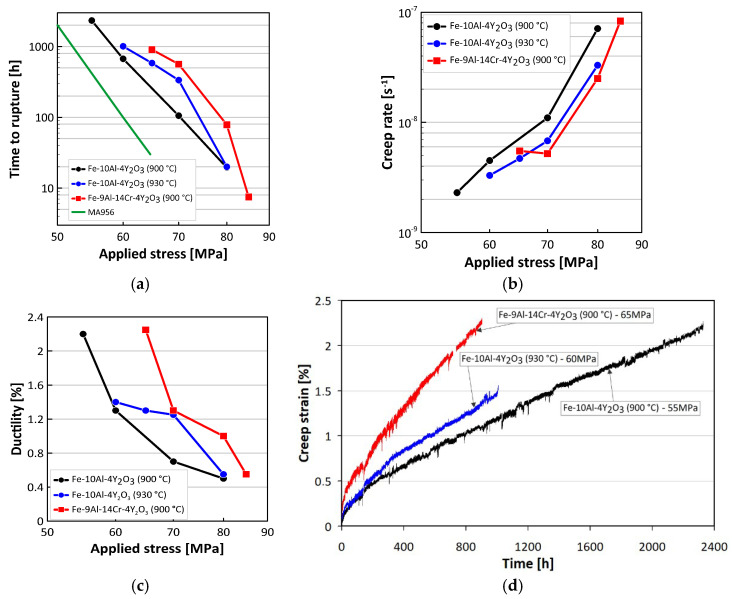
Summarizing results creep at 1100 °C of the most promising heats Fe-10Al-4Y_2_O_3_ rolled at 900 °C and 930 °C and Fe-9Al-14Cr-4Y_2_O_3_ rolled at 930 °C, (**a**) dependences of time to fracture on applied stress, (**b**) dependences of secondary creep rate on applied stress, (**c**) dependences of ductility on applied stress, and (**d**) longest creep curve of the studied heats.

**Figure 7 materials-13-05070-f007:**
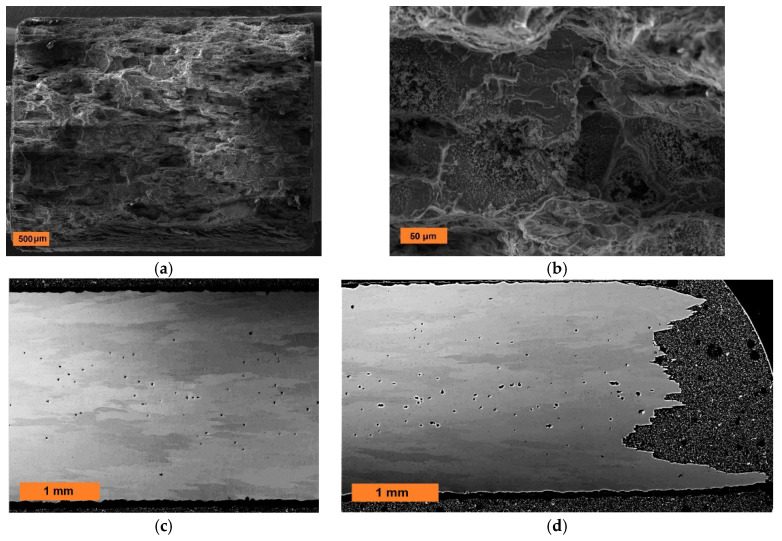
The fracture analysis of the heat Fe-10Al-4Y_2_O_3_ crept at 55 MPa. (**a**) The overall view on the fracture surface, (**b**) detail of intergranular cavities, (**c**) porosity in the head of the specimen, and (**d**) porosity near the fracture.

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
