# Peer review of "Influence of Hot Consolidation Conditions and Cr-Alloying on Microstructure and Creep in New-Generation ODS Alloy at 1100 °C"

_materials, 2020, doi:10.3390/ma13225070_

Round 1
Reviewer 1 Report
- Fig. 2b should be indicated as a TEM micrograph. In addition, the scale bar does not appear to be correct.
- Although it is optional, an analysis of the texture and its effect on mechanical properties would be good, given that the alloys were processed by hot rolling,.
- As the authors stated, it is pity that there exist quite a few gas pores in the alloys after hot rolling, which should have detrimental effects on mechanical properties and creep resistance. To prevent the gas pores, the MA powder may need to be degassed and canned before hot rolling. This is apparently out of the scope of this paper.
Author Response
Reviewer 1
We would like to thank Reviewer 1 for her/his comments that improve the quality of the paper.
- Fig. 2b should be indicated as a TEM micrograph. In addition, the scale bar does not appear to be correct.
Response: The TEM images are published in our previous paper with reference no. [29], which was referenced incorrectly, because references [29] and [31] had to be exchanged (corrected in the revised version).
- Although it is optional, an analysis of the texture and its effect on mechanical properties would be good, given that the alloys were processed by hot rolling,.
Response: The texture was analyzed in the paper just submitted to Materials, we added the reference [30].
- As the authors stated, it is pity that there exist quite a few gas pores in the alloys after hot rolling, which should have detrimental effects on mechanical properties and creep resistance. To prevent the gas pores, the MA powder may need to be degassed and canned before hot rolling. This is apparently out of the scope of this paper.
Response: We thank for this pertinent comment. Currently, we are concentrating on the powder degassing issues, which, as we see now, deserves very detailed analysis. A paper dealing with that problem will be published later. Some comments are added in the revised version.
Reviewer 2 Report
The manuscript is attempting to present a comparison of the influence of rolling temperature and Cr-alloying on the microstructure and creep properties of two developed ODS alloys, as stated in the objective statements in lines 76-79 of the paper. While the authors appear to make strides in presenting these comparisons (which appear to be meaningful) some aspects of the manuscript appear to not be adequately supported. Some suggestions to make the manuscript more robust are:
1) Within the Introduction, there are numerous mentions of prior work by the author, but no references provided. The requires the reader to accept the prior work without any opportunity to review it. Examples are in lines 66 and lines 67-75.
2) The Materials and Methods section seems to be continually mixing up the methodology and a partial summary of the results, which such results are not substantiated with any insight into how they were observed. For example, in line 92 there is mention of small amounts of Cr, but no mention of how much Cr or how it was measured/observed. Shouldn’t any objective measurements belong in the results. Another example is in lines 100-101, reference ultra-fine grained pore free microstructure. This sounds more like a “Result”, and there is no mention of how this was determined, or any reference to prior work that confirms this. Therefore, how do you know this is the case?
3) The Microstructure characterization section appears to be purely subjective in nature. Imaging is conducted, but no objective measurements of the microstructure are included. The arguments about observe grain sizes in Fig. 3 and 4 would be much stronger if objective grain size measurements were conducted to quantify this. Line 156 references a “mean size”, yet no quantifiable data is provided.
4) The statement in lines 142-143 needs to be supported with references.
5) Lines 143-145 discuss coarsening of the oxides to 20 nm. It in not clear if this is what you observed, or just a statement about the evolution of the pinning effect. If this was observed, it needs to be clearly stated, and you need to indicate how you measured this.
6) The conclusions section is not very strong:
6a. Lines 284 – 286 does not appear to be a concluding statement, but instead just recaps what you did (?).
6b. Line 289 – how did you determine the values of 20 nm and 5%? The manuscript makes no mention of how this is measured.
6c. Line 287-288 would be stronger if you actually explained the trends and why they are important.
6d. There is no specific conclusion regarding the Cr-alloying influence, despite the fact that this was one of the original objectives of the manuscript, as stated in line 78-79.
Author Response
We would like to thank Reviewer 2 for her/his comments that improve the quality of the paper.
The manuscript is attempting to present a comparison of the influence of rolling temperature and Cr-alloying on the microstructure and creep properties of two developed ODS alloys, as stated in the objective statements in lines 76-79 of the paper. While the authors appear to make strides in presenting these comparisons (which appear to be meaningful) some aspects of the manuscript appear to not be adequately supported. Some suggestions to make the manuscript more robust are:
1) Within the Introduction, there are numerous mentions of prior work by the author, but no references provided. The requires the reader to accept the prior work without any opportunity to review it. Examples are in lines 66 and lines 67-75.
Response: We added references and reformulated the respective text in the revised manuscript.
2) The Materials and Methods section seems to be continually mixing up the methodology and a partial summary of the results, which such results are not substantiated with any insight into how they were observed. For example, in line 92 there is mention of small amounts of Cr, but no mention of how much Cr or how it was measured/observed. Shouldn’t any objective measurements belong in the results. Another example is in lines 100-101, reference ultra-fine grained pore free microstructure. This sounds more like a “Result”, and there is no mention of how this was determined, or any reference to prior work that confirms this. Therefore, how do you know this is the case?
Response: We added the respective reference and explanations in the revised manuscript.
3) The Microstructure characterization section appears to be purely subjective in nature. Imaging is conducted, but no objective measurements of the microstructure are included. The arguments about observe grain sizes in Fig. 3 and 4 would be much stronger if objective grain size measurements were conducted to quantify this. Line 156 references a “mean size”, yet no quantifiable data is provided.
Response: We added the values of mean grain sizes for the individual grain microstructures.
4) The statement in lines 142-143 needs to be supported with references.
Response: We are apologizing for the incorrect reference, which is now corrected in the revised manuscript.
5) Lines 143-145 discuss coarsening of the oxides to 20 nm. It in not clear if this is what you observed, or just a statement about the evolution of the pinning effect. If this was observed, it needs to be clearly stated, and you need to indicate how you measured this.
Response: We are apologizing again. This is supported by the thermodynamic model [29] (incorrectly referenced in the original manuscript), where the simulations based on the model support our statements.
6) The conclusions section is not very strong:
6a. Lines 284 – 286 does not appear to be a concluding statement, but instead just recaps what you did (?).
Response: We agree with the reviewer. We shifted this part into the introducing sentence in this section.
6b. Line 289 – how did you determine the values of 20 nm and 5%? The manuscript makes no mention of how this is measured.
Response: We added "From the Figure 2b one can estimate the mean size of the precipitates to be 20 nm and their volume fraction of 5 % follows from the chemical composition and the assumption that all oxygen is bond in the yttrium oxides."
6c. Line 287-288 would be stronger if you actually explained the trends and why they are important.
Response: We have stressed that the trends (influence of rolling temperature and chemical composition) are not very significant for the final microstructure.
6d. There is no specific conclusion regarding the Cr-alloying influence, despite the fact that this was one of the original objectives of the manuscript, as stated in line 78-79.
Response: We have added one item in the conclusions.
Reviewer 3 Report
First of all, I have to say that the paper is really interesting and the authors have a good knowledge of the subject.
I have some things are not clear:
- Figure 2.b: how can you say with this SEM micrograph the secondary particles have 5nm?
- The SEM micrographs 3 and 4, I do not understand for the text if the material is annealing at high temperature or not.
- Figure 4, the authors identify the Cr particles. Where is the analysis for this statement?
Author Response
We would like to thank Reviewer 3 for her/his comments that improve the quality of the paper.
First of all, I have to say that the paper is really interesting and the authors have a good knowledge of the subject.
I have some things are not clear:
Figure 2.b: how can you say with this SEM micrograph the secondary particles have 5nm?
Response: Oxides after secondary recrystallization that have average size 20nm are shown in Figure 2b. The 5 nm oxides after hot consolidation are shown in Ref. [29] Fig. 1b (TEM image).
The SEM micrographs 3 and 4, I do not understand for the text if the material is annealing at high temperature or not.
Response: Yes the micrographs are after secondary recrystallization (annealing) at 1200°C/4h, the statement has been added.
Figure 4, the authors identify the Cr particles. Where is the analysis for this statement?
Response: The detected carbides are the mixed (Fe,Cr)3C carbides as it is added in the text.
Round 2
Reviewer 2 Report
The authors appear to have adequately addressed each of the Reviewer's comments.
Reviewer 3 Report
Thanks for taking in account my suggestions
It will be interesting that you include the in the EDS analysis in Figure 4.